# Risk of myocarditis and pericarditis after a COVID-19 mRNA vaccine booster and after COVID-19 in those with and without prior SARS-CoV-2 infection: A self-controlled case series analysis in England

Julia Stowe[1], Elizabeth Miller[2]*, Nick Andrews[1], Heather J. Whitaker[1]

**1** UK Health Security Agency, London, United Kingdom, **2** NIHR Health Protection Research Unit in Vaccines and Immunisation, London School of Hygiene and Tropical Medicine, London, United Kingdom

* liz.miller@lshtm.ac.uk

**Data Availability Statement:** The raw study data are protected and are not freely available due to data privacy laws. This work is carried out under Regulation 3 of The Health Service (Control of Patient Information) (Secretary of State for Health, 2002))(3) using patient identification information

## Abstract

### Background

An increased risk of myocarditis or pericarditis after priming with mRNA Coronavirus Disease 2019 (COVID-19) vaccines has been shown but information on the risk post-booster is limited. With the now high prevalence of prior Severe Acute Respiratory Syndrome Coronavirus 2 (SARS-CoV-2) infection, we assessed the effect of prior infection on the vaccine risk and the risk from COVID-19 reinfection.

### Methods and findings

We conducted a self-controlled case series analysis of hospital admissions for myocarditis or pericarditis in England between 22 February 2021 and 6 February 2022 in the 50 million individuals eligible to receive the adenovirus-vectored vaccine (ChAdOx1-S) for priming or an mRNA vaccine (BNT162b2 or mRNA-1273) for priming or boosting. Myocarditis and pericarditis admissions were extracted from the Secondary Uses Service (SUS) database in England and vaccination histories from the National Immunisation Management System (NIMS); prior infections were obtained from the UK Health Security Agency's Second-Generation Surveillance Systems. The relative incidence (RI) of admission within 0 to 6 and 7 to 14 days of vaccination compared with periods outside these risk windows stratified by age, dose, and prior SARS-CoV-2 infection for individuals aged 12 to 101 years was estimated. The RI within 27 days of an infection was assessed in the same model. There were 2,284 admissions for myocarditis and 1,651 for pericarditis in the study period. Elevated RIs were only observed in 16- to 39-year-olds 0 to 6 days postvaccination, mainly in males for myocarditis. Both mRNA vaccines showed elevated RIs after first, second, and third doses with the highest RIs after a second dose 5.34 (95% confidence interval (CI) [3.81, 7.48]; $p < 0.001$) for BNT162b2 and 56.48 (95% CI [33.95, 93.97]; $p < 0.001$) for mRNA-1273 compared with 4.38 (95% CI [2.59, 7.38]; $p < 0.001$) and 7.88 (95% CI [4.02, 15.44]; $p < 0.001$),

without individual patient consent. Data cannot be made publicly available for ethical and legal reasons, i.e. public availability would compromise patient confidentiality as data tables list single counts of individuals rather than aggregated data. Requests for the underlying data should be made via the UKHSA office for data release: https://www.gov.uk/government/publications/accessing-ukhsa-protected-data.

**Funding:** This work was supported by the UK Health Security Agency for authors NA, JS HJW via their employment. EM receives support from the National Institute for Health Research Health Protection Research Unit in Immunisation at the London School of Hygiene and Tropical Medicine in partnership with UKHSA (Grant Reference NIHR200929). The funders had no role in study design, data collection and analysis, decision to publish, or preparation of the manuscript.

**Competing interests:** The authors have declared that no competing interests exist.

**Abbreviations:** CEV, clinically extremely vulnerable; CI, confidence interval; COVID-19, Coronavirus Disease 2019; dsRNA, double-stranded RNA; ECDS, Emergency Care Data Set; NHS, National Health Service; NIMS, National Immunisation Management System; RI, relative incidence; SAE, serious adverse event; SARS-CoV-2, Severe Acute Respiratory Syndrome Coronavirus 2; SCCS, self-controlled case series; SGSS, Second Generation Surveillance System; SUS, Secondary Uses Service.

respectively, after a third dose. For ChAdOx1-S, an elevated RI was only observed after a first dose, RI 5.23 (95% CI [2.48, 11.01]; $p < 0.001$). An elevated risk of admission for pericarditis was only observed 0 to 6 days after a second dose of mRNA-1273 vaccine in 16 to 39 year olds, RI 4.84 (95% CI [1.62, 14.01]; $p = 0.004$). RIs were lower in those with a prior SARS-CoV-2 infection than in those without, 2.47 (95% CI [1.32,4.63]; $p = 0.005$) versus 4.45 (95% [3.12, 6.34]; $p = 0.001$) after a second BNT162b2 dose, and 19.07 (95% CI [8.62, 42.19]; $p < 0.001$) versus 37.2 (95% CI [22.18, 62.38]; $p < 0.001$) for mRNA-1273 (myocarditis and pericarditis outcomes combined). RIs 1 to 27 days postinfection were elevated in all ages and were marginally lower for breakthrough infections, 2.33 (95% CI [1.96, 2.76]; $p < 0.001$) compared with 3.32 (95% CI [2.54, 4.33]; $p < 0.001$) in vaccine-naïve individuals respectively.

## Conclusions

We observed an increased risk of myocarditis within the first week after priming and booster doses of mRNA vaccines, predominantly in males under 40 years with the highest risks after a second dose. The risk difference between the second and the third doses was particularly marked for the mRNA-1273 vaccine that contains half the amount of mRNA when used for boosting than priming. The lower risk in those with prior SARS-CoV-2 infection, and lack of an enhanced effect post-booster, does not suggest a spike-directed immune mechanism. Research to understand the mechanism of vaccine-associated myocarditis and to document the risk with bivalent mRNA vaccines is warranted.

## Author summary

### Why was this study done?

- Primary and booster immunisation with mRNA Coronavirus Disease 2019 (COVID-19) vaccine have been associated with an increased risk of acute myocarditis.

- Severe Acute Respiratory Syndrome Coronavirus 2 (SARS-CoV-2) infection may itself cause myocarditis or pericarditis.

- However, the effect of prior vaccination on this risk, and on the risk after a reinfection, has not been investigated.

### What did the researchers do and find?

- We conducted a nationwide study in England to assess the risk of hospital admission for myocarditis or pericarditis after primary or booster and the risk after a confirmed SARS-CoV-2 infection in those with and without a prior confirmed SARS-CoV-2 infection.

- Elevated risks of myocarditis were found up to 6 days after each of priming dose of the available mRNA vaccines (BNT162b2 and mRNA-1723) and after mRNA booster doses following a mRNA priming course but not after a priming course of the adenovirus-

vectored vaccine ChAdOx1-S. The only elevated seen after the ChAdOx1-S vaccine was after the dose in 16 to 39 year olds.

- For both mRNA vaccines, elevated risks were found in those under 40 years old, predominantly in males, were highest after the second priming dose and were generally lower in those vaccinated after a prior SARS-CoV-2 infection.

- There was an elevated risk of myocarditis and pericarditis in the 27 days after a SARS-CoV-2 infection which was higher in ≥40 year olds than 16 to 39 year olds and was still present in those with a reinfection or who had been vaccinated before infection.

### What do these findings mean?

- This study provides information for policy makers and those recommended to receive booster mRNA vaccines on the associated rare risk of myocarditis or pericarditis in a population with a high prevalence of prior SARS-CoV-2 infection.

- The lower risk after a booster than primary course, and the lower risk in vaccinees with a prior SARS-CoV-2 infection, does not suggest an immune-mediated mechanism directed at the spike protein.

- The greater risk associated with mRNA-1273 vaccines, which have a higher mRNA dose than BNT162b2 vaccines, and the substantially lower risk after the mRNA-booster which has half the mRNA content than used for priming, may be suggestive an mRNA dose-related mechanism but further work is required to determine this.

## Introduction

The rapid development and global deployment of Coronavirus Disease 2019 (COVID-19) vaccines based on mRNA technology has been one of the outstanding successes of the response to the Severe Acute Respiratory Syndrome Coronavirus 2 (SARS-CoV-2) pandemic. Licensed for human use for the first time in the pandemic, mRNA vaccines have proven highly effective at preventing severe morbidity and mortality from SARS-CoV-2 infection not only against the original Wuhan strain but subsequent variants of concern [1,2]. Overall, their safety profile has been good with no serious adverse events (SAEs) detected until the reports from Israel of temporally associated cases of acute myocarditis and pericarditis after primary vaccination with the BNT162b2 vaccine [3]. Reported cases were predominantly in younger males after the second dose, with onset clustering in the first week after vaccination.

Subsequent epidemiological studies in Israel and other countries using BNT162b2 and the mRNA-1273 vaccine confirmed an increased risk of myocarditis, largely after the second dose [4–8]. However, limited use of adenovirus-vectored vaccines such as ChAdOx1-S in these settings precluded a comparison of the risk by type of vaccine platform. Furthermore, these studies were conducted before the introduction of booster doses of COVID-19 vaccines, now recommended in many high-income settings where mRNA and adenovirus-vectored vaccines have been deployed. While there are case reports of acute myocarditis after booster doses of BNT162b2 vaccine, there is a paucity of epidemiological studies evaluating the risk after a booster dose of mRNA vaccines [9–11].

In the United Kingdom, there has been widespread use of the 2 mRNA vaccines and ChAdOx1-S vaccine with high population coverage both for the primary 2 dose course and the booster dose, the latter restricted to mRNA vaccines. A self-controlled case series (SCCS) analysis of the risk of hospital admission for myocarditis or pericarditis in England in those aged 16 years and older found an elevated risk within 28 days of a first dose of ChAdOx1-S and mRNA-1273 vaccines and after the second dose for each of the 2 mRNA vaccines [12]. This analysis was subsequently updated to mid-December 2021 to capture early data from the booster programme that began in mid-September 2021 with the BNT162b2 vaccine and was targeted at older age adults. While there was evidence of an elevated risk within 28 days of a booster dose of BNT162b2 vaccine, there was inadequate power to assess the risk by age and sex, nor for the mRNA-1273 vaccine [13]. We estimate the risk of myocarditis or pericarditis after a booster dose of BNT162b2 or mRNA-1273 vaccines in England by age and sex. We stratify post-booster outcomes according to the vaccine platform used for priming and assess the effect of prior SARS-CoV-2 infection on the vaccine-associated risk. We also assess the risk after a first or subsequent SARS-CoV-2 infection and the risk after a breakthrough infection in vaccinees.

## Methods

### Study population and study period

The study population comprised the resident population of 50 million individuals in England aged 12 years and older on the 31 August 2021. Age was defined as of 31 August 2021 as this best reflected eligibility for vaccination in the paediatric programme for 12 to 15 year olds that started after this date. Dates of admission for myocarditis or pericarditis were from 22 February 2021 to 6 February 2022.

### Study design

Given the generally mild nature of vaccine-associated myocarditis [14], we analysed cases presenting in emergency care settings as well as those admitted to hospital. Two analytic methods were used, an SCCS analysis [15] supplemented by a retrospective cohort analysis. Both analyses assessed whether there was an increased risk of presentation to emergency care or admission to hospital with myocarditis and/or pericarditis in prespecified risk periods after any of the 3 COVID-19 vaccines used in England; ChAdOx1-S, BNT162b2, or mRNA-1273 vaccines. The SCCS analysis assessed the risk separately for myocarditis and pericarditis. The risk of acute myocarditis and/or pericarditis after a confirmed SARS-CoV-2 infection was also assessed in the SCCS analysis. All analyses used date of admission/attendance as the index date.

### Vaccination database

Immunisation data was obtained from the National Immunisation Management System (NIMS), an individual level centralised register for the management of both seasonal influenza and COVID-19 vaccination records across England. It comprises the demographic characteristics for all residents in England eligible to receive COVID-19 vaccination (approximately 50 million individuals) including whether the person is in a priority group for COVID-19 vaccination because of comorbidities considered to render the individual clinically extremely vulnerable (CEV) [16]. A wider group of individuals with other comorbidities were flagged in the NIMS database retrospectively in mid-February 2021, hence the restriction of the start of the study period to late February 2021.

### Emergency Care Data Set (ECDS)

Emergency care hospital attendances for myocarditis and/or pericarditis from the Emergency Care Data Set (ECDS), which is the National Health Service (NHS) dataset for urgent and emergency care in England. The attendances were identified using SNOMED CT (Systematized Nomenclature of Medicine–Clinical Terms) codes 50920009 myocarditis or 3238004 pericarditis. Only the first ECDS consultation during the study period was included in those without a prior ECDS consultation since 1 December 2019. The extracted ECDS attendances with the outcomes of interest that did not link with an NIMS record were excluded from the analysis; these comprised 0.8% of the extracted ECDS attendances.

### Hospital admissions database (SUS)

The Secondary Uses Service (SUS) dataset, a database of timely completed hospital admissions for all NHS hospitals in England, was used to identify an individual's first admission due to myocarditis and/or pericarditis in the study period (with no prior admission since 1 December 2019) using ICD10 codes I30 acute pericarditis, I40 acute myocarditis, and I51.4 myocarditis, unspecified in the first 3 diagnosis fields. The extracted SUS admissions with the outcomes of interest that did not link with an NIMS record were excluded from the analysis; these comprised 0.6% of the extracted SUS admissions.

### SARS-CoV-2 infection dataset

The results of PCR and LFT tests carried out in the community in England, and PCR tests conducted in hospital patients, are collated in the UKHSA Second Generation Surveillance System (SGSS) that was used to identify confirmed SARS-CoV-2 infections in the study population. Repeat infections were defined as those in individuals with a second or subsequent positive test ≥90 days after a previous positive test.

### Data linkage

Emergency care attendances, hospital admission records, and SGSS SARS-CoV-2 cases were linked to the NIMS dataset using NHS number. In all datasets used in the analysis, the NHS numbers were checked as valid using the final digit checksum that detects erroneous NHS numbers, which could lead to invalid linkage.

### Construction of the self-controlled case series dataset (SCCS)

The SCCS dataset comprised of individuals aged 12 years and above with an ECDS consultation or hospital admission for myocarditis and/or pericarditis in the study period and who had received at least 1 dose of COVID-19 vaccine or had 1 or more confirmed SARS-CoV-2 infections at least 90 days apart. Vaccinated individuals who had received a mixed primary schedule, first and second doses <19 days apart, or second and third doses <56 days apart, or a third ChAdOx1-S dose, or a third dose before 1 September 2021 (before the booster programme started), or a first dose before 8 December 2020 (before the national vaccination programme started) were excluded as were any recipients of other COVID-19 vaccines that may have been given as part of a vaccine trial.

### Construction of the cohort study dataset

NIMS monthly denominator and daily vaccination data were used to construct the cumulative vaccination status of the population eligible to receive COVID-19 vaccination by day stratified by age, gender, vaccine type, postvaccination intervals, CEV and other clinical risk group,

ethnic group and English region. Postvaccination intervals were stratified by 0 to 6, 7 to 13, 14 + days after a first, second dose or third dose, and separately for ChAdOx1-S, BNT162b2, and mRNA-1273 vaccines (and for these primary schedules combined with the BNT162b2 and mRNA 1273 boosters), or unvaccinated. The same exclusions were applied to mixed or other nonstandard schedules as in the SCCS dataset. Age was stratified into 12 to 15, 16 to 17, 18 to 19, then 5-year bands. Ethnic group was collapsed into 5 main groups: White, Mixed or Multiple ethnic groups, Asian or Asian British, Black, African, Caribbean or Black British, and an unknown group.

Myocarditis and pericarditis events were stratified by the same factors and were merged with the NIMS data to obtain a stratified dataset of event counts and population denominators by day during the study period. Individuals who were initially unvaccinated but then received a vaccine in the study period, contributed to the unvaccinated person time. As outcome events were rare, for computational simplicity and to allow more rapid delivery of results for policy makers person time was censored at 6 February 2022 not at event date. Individuals who died were removed from the cohort at the end of the month in which they died using information on deaths in the NIMS denominator files.

## Statistical analysis plan

A statistical analysis plan was drawn up in advance as part of the protocol (S1 Protocol) for which the key elements are summarised below.

## SCCS analysis

The relative incidence (RI) of myocarditis and/or pericarditis in specified risk periods after vaccination and SARS-CoV-2 infection were estimated in the same model. A key assumption of the SCCS method is that the exposure and event are independent—an assumption which is violated if vaccination is deferred after an event until recovery (short-term event dependence), or subsequent doses are contraindicated after an event (long-term event dependence). The model accounted for short-term dependence, with the appropriate length of the preexposure period investigated; long-term event dependence was also investigated (S1 Appendix). These analyses indicated a pre-vaccination period of 21 days would be adequate to account for short-term event dependence and that there was no major concerns regarding longer-term dependence.

For the vaccination effect, the incidence in the 0 to 6 and 7 to 13 days after any dose of COVID-19 vaccine was compared to the incidence periods in vaccinated individuals outside this window using a dataset restricted to vaccinated individuals with myocarditis or pericarditis. Period adjustment in 4 weekly intervals was included in the model. Analyses comprised (i) all ages 12+; (ii) restricting to ages 16 to 39; (iii) ages 40+; (iv) ages 12 to 15; (v) males only; and (vi) females only. The analyses in ages 12+, 16 to 39, and 40+ in SUS using the SCCS method and with myocarditis and pericarditis combined were the primary analyses. As there were 16 vaccine dose and post vaccine interval combinations for each of these, this gave a total of 48 tests. All other analyses were secondary or exploratory. To account for this large number of primary and secondary assessments, the prespecified significance level was set at $p < 0.001$. RI was not estimated for risk periods with <2 cases. P-values were calculated using the Z test. In an exploratory analysis, vaccinated individuals were restricted to those with a prior confirmed SARS-CoV-2 infection to investigate whether preexisting immunity affected the vaccine risk. To assess the potential for ascertainment bias when the association between mRNA vaccination and myocarditis was publicised by the Medical and Healthcare products Agency (MHRA), a sensitivity analysis restricting cases to those presenting by 23 August 2021 was also

conducted. At the request of reviewers further stratification of myocarditis results for those aged 16 to 39 years by age and gender were conducted.

For assessing the risk of SARS-CoV-2 infection, the exposure date was the first positive test date for that individual in the study period plus any subsequent new positive tests separated by ≥90 days. A postinfection risk period of 1 to 27 days was specified with those tested on the day they present to hospital or emergency care (day 0) analysed separately. A 14-day preinfection window was included. Two datasets were analysed, one restricted to vaccinated individuals and the other including in addition unvaccinated individuals with a confirmed SARS-CoV-2 infection.

In ad hoc analyses, the infection risk was stratified by first infection or reinfection, vaccine status (unvaccinated or at least 1 dose), and by variant using date of infection as a proxy for infecting variant (Delta replacing Alpha on 17 May and Omicron replacing Delta on 13 December).

## Cohort analysis

Poisson regression was used to estimate the RI of events in the prespecified postvaccination risk periods compared with the unvaccinated period with an offset for population at risk (person days). The model adjusted for age group (12 to 15, 16 to 17, 18 to 19, then 5-year bands), gender, ethnic group, region, CEV, other clinical risk group, and 4-week interval. The core model had an age stratification of 12 to 15, 16 to 39, 40+, as well as all ages combined, with additional analyses showing stratification by gender (all ages).

## Attributable risk estimates

Attributable cases in the risk intervals with RIs $p < 0.001$ were estimated from the attributable fraction AF = (RI-1)/RI multiplied by the number of cases in that interval. Attributable risk was then calculated from the attributable cases divided by either the number of doses administered (for vaccine risk) or the number of SARS-CoV-2 infections or reinfections (for SARS-CoV-2 infection risk).

## STROBE guidelines

The study is reported as per the Strengthening the Reporting of Observational Studies in Epidemiology (STROBE) guideline (S1 STROBE Checklist).

## Results

There was a total of 3,124 hospital admissions in England with a diagnosis of myocarditis or pericarditis and 7,933 emergency care consultations between 22 February 2021 and 6 February 2022 (Table 1). Admission rates for myocarditis were generally higher than for pericarditis, whereas ECDS consultation rates were higher for pericarditis than myocarditis, particularly among those aged under 40 years. Confirmed SARS-CoV-2 infections in those admitted with myocarditis or pericarditis reflected the changing incidence of COVID-19 over the study period with a drop in the last 4 weeks due to delays in diagnostic coding and admissions not completed by the time of the data extract on 4 April 2022 (Fig 1). At least 1 dose of a COVID-19 vaccine had been received by 6,672 (84.1%) of those attending emergency care and 3,382 (86.2%) of those admitted to hospital. In both the ECDS and SUS datasets, admission rates per 100,000 person years in males were about double those in females and increased sharply with age between 12 and 19 years, remaining fairly constant thereafter. ECDS consultation and SUS

**Table 1.** Demographic and clinical features of the individuals with a hospital admission in SUS or an emergency care consultation in the ECDS dataset for myocarditis and pericarditis: data from entire eligible population in England.

| | | person years | SUS pericarditis case count = 1,651* | SUS pericarditis risk per 100,000 | SUS myocarditis case count = 2,284* | SUS myocarditis risk per 100,000 | ECDS pericarditis case count = 6,461 | ECDS pericarditis risk per 100,000 | ECDS myocarditis case count = 1,472 | ECDS myocarditis risk per 100,000 |
|---|---|---|---|---|---|---|---|---|---|---|
| Age group | 12 to 15 | 2,742,347 | 8 | 0.29 | 28 | 1.02 | 111 | 4.05 | 42 | 1.53 |
| | 16 to 17 | 1,305,334 | 30 | 2.30 | 73 | 5.59 | 151 | 11.57 | 71 | 5.44 |
| | 18 to 19 | 1,296,676 | 54 | 4.16 | 119 | 9.18 | 241 | 18.59 | 118 | 9.10 |
| | 20 to 24 | 3,730,389 | 119 | 3.19 | 233 | 6.25 | 757 | 20.29 | 241 | 6.46 |
| | 25 to 29 | 4,196,722 | 117 | 2.79 | 199 | 4.74 | 754 | 17.97 | 184 | 4.38 |
| | 30 to 34 | 4,513,956 | 127 | 2.81 | 200 | 4.43 | 715 | 15.84 | 163 | 3.61 |
| | 35 to 39 | 4,297,665 | 106 | 2.47 | 166 | 3.86 | 705 | 16.40 | 144 | 3.35 |
| | 40 to 44 | 3,956,356 | 126 | 3.18 | 153 | 3.87 | 565 | 14.28 | 91 | 2.30 |
| | 45 to 49 | 3,768,292 | 131 | 3.48 | 155 | 4.11 | 517 | 13.72 | 84 | 2.23 |
| | 50 to 54 | 4,013,470 | 140 | 3.49 | 157 | 3.91 | 486 | 12.11 | 61 | 1.52 |
| | 55 to 59 | 3,903,178 | 139 | 3.56 | 166 | 4.25 | 431 | 11.04 | 74 | 1.90 |
| | 60 to 64 | 3,320,364 | 124 | 3.73 | 145 | 4.37 | 293 | 8.82 | 57 | 1.72 |
| | 65 to 69 | 2,773,874 | 111 | 4.00 | 113 | 4.07 | 242 | 8.72 | 45 | 1.62 |
| | 70 to 74 | 2,732,488 | 125 | 4.57 | 122 | 4.46 | 191 | 6.99 | 36 | 1.32 |
| | 75 to 79 | 2,069,915 | 82 | 3.96 | 101 | 4.88 | 146 | 7.05 | 31 | 1.50 |
| | 80 to 84 | 1,390,802 | 61 | 4.39 | 91 | 6.54 | 83 | 5.97 | 17 | 1.22 |
| | 85 to 89 | 868,675 | 39 | 4.49 | 44 | 5.07 | 55 | 6.33 | 10 | 1.15 |
| | 90+ | 504,934 | 12 | 2.38 | 19 | 3.76 | 18 | 3.56 | 3 | 0.59 |
| Sex | Male | 25,717,812 | 1,132 | 4.40 | 1,456 | 5.66 | 4,634 | 18.02 | 1,001 | 3.89 |
| | Female | 25,667,624 | 519 | 2.02 | 828 | 3.23 | 1,827 | 7.12 | 471 | 1.83 |
| Region | East of England | 5,910,664 | 175 | 2.96 | 335 | 5.67 | 703 | 11.89 | 165 | 2.79 |
| | London | 8,807,901 | 249 | 2.83 | 420 | 4.77 | 973 | 11.05 | 239 | 2.71 |
| | Midlands | 9,496,410 | 220 | 2.32 | 321 | 3.38 | 986 | 10.38 | 248 | 2.61 |
| | North East and Yorkshire | 7,641,777 | 217 | 2.84 | 273 | 3.57 | 938 | 12.27 | 186 | 2.43 |
| | North West | 6,393,140 | 215 | 3.36 | 278 | 4.35 | 713 | 11.15 | 199 | 3.11 |
| | South East | 8,025,838 | 365 | 4.55 | 445 | 5.54 | 1,230 | 15.33 | 267 | 3.33 |
| | South West | 5,109,705 | 210 | 4.11 | 212 | 4.15 | 859 | 16.81 | 153 | 2.99 |
| Ethnicity | Asian | 4,398,660 | 94 | 2.14 | 155 | 3.52 | 513 | 11.66 | 103 | 2.34 |
| | Black, African, Caribbean | 1,756,382 | 108 | 6.15 | 133 | 7.57 | 468 | 26.65 | 95 | 5.41 |
| | Mixed, Multiple | 820,496 | 22 | 2.68 | 45 | 5.48 | 145 | 17.67 | 38 | 4.63 |
| | NK | 6,752,624 | 95 | 1.41 | 182 | 2.70 | 448 | 6.63 | 101 | 1.50 |
| | Other | 1,173,462 | 42 | 3.58 | 69 | 5.88 | 165 | 14.06 | 48 | 4.09 |
| | White | 36,483,810 | 1,290 | 3.54 | 1,700 | 4.66 | 4,722 | 12.94 | 1,087 | 2.98 |
| Clinically vulnerable | No | 47,787,906 | 1,387 | 2.90 | 1,943 | 4.07 | 5,830 | 12.20 | 1,333 | 2.79 |
| | Yes | 3,597,530 | 264 | 7.34 | 341 | 9.48 | 631 | 17.54 | 139 | 3.86 |
| In an "at risk" group | No | 43,400,928 | 1,107 | 2.55 | 1,446 | 3.33 | 4,545 | 10.47 | 1,010 | 2.33 |
| | Yes | 7,984,508 | 544 | 6.81 | 838 | 10.50 | 1,916 | 24.00 | 462 | 5.79 |
| Unvaccinated | | 16,751,085 | 206 | 1.23 | 340 | 2.03 | 1,036 | 6.18 | 225 | 1.34 |

*(Continued)*

**Table 1.** (Continued)

| | | person years | SUS pericarditis case count = 1,651* | SUS pericarditis risk per 100,000 | SUS myocarditis case count = 2,284* | SUS myocarditis risk per 100,000 | ECDS pericarditis case count = 6,461 | ECDS pericarditis risk per 100,000 | ECDS myocarditis case count = 1,472 | ECDS myocarditis risk per 100,000 |
|---|---|---|---|---|---|---|---|---|---|---|
| **Primary course** | ChAdOx1 | 18,381,521 | 728 | 3.96 | 859 | 4.67 | 2,332 | 12.69 | 439 | 2.39 |
| | BNT162b2 | 15,459,412 | 670 | 4.33 | 972 | 6.29 | 2,833 | 18.33 | 728 | 4.71 |
| | mRNA-1273 | 793,417 | 47 | 5.92 | 112 | 14.12 | 259 | 32.64 | 79 | 9.96 |

*Eleven SUS cases had a diagnosis of both myocarditis and pericarditis.

ECDS, Emergency Care Data Set; SUS, Secondary Uses Service.

admission rates were higher among black British, African, or Caribbean than other ethnic groups and higher among those with a CEV or other "at risk" flag than in those without.

## Postvaccination analyses

Elevated RI estimates ($p < 0.01$ or $<0.001$) for hospital admission with myocarditis after vaccination were found in the SCCS analysis in those aged under 40 years in the 0 to 6 day postvaccination period after each of the mRNA vaccines and also after the first dose of ChAdOx1-S vaccine (Table 2). RIs were highest after the second mRNA dose and generally higher after the mRNA-1273 than the BNT162b2 vaccine, though more similar when given as a booster. Stratification of the results by sex showed elevated RIs only in males with the exception of 0 to 6 days after a second dose of mRNA-1273 vaccine for which the RI was 58.28 (95% confidence interval (CI) [34.78, 97.52], $p < 0.001$) in males and 23.26 (95% CI [6.68, 82.07], $p < 0.001$) in females (Table 2).

Further age stratification within 16 to 39 year olds showed higher risks in 16 to 24 than 25 to 39 year olds, RI 102.3 (95% CI [49.88, 209.6], $p < 0.001$) compared with 29.67 (95% CI

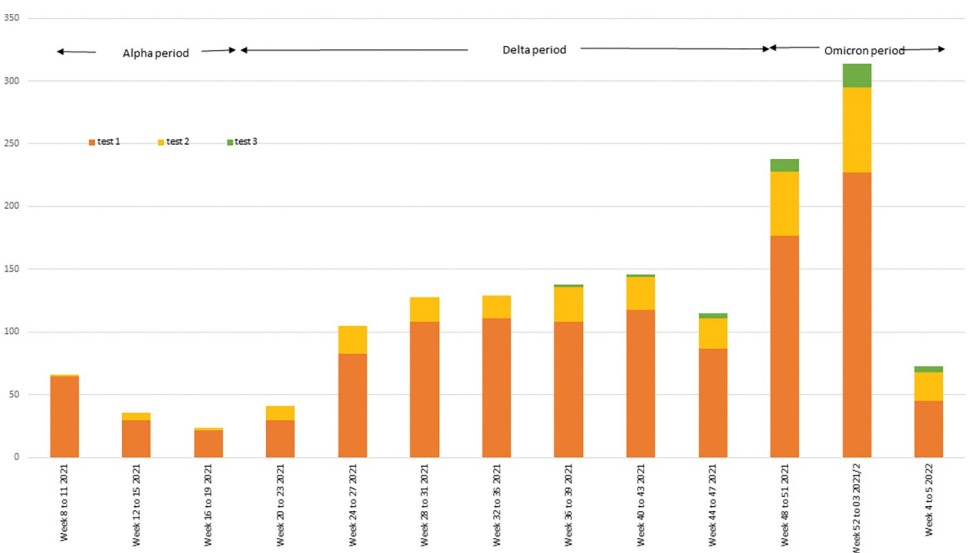

**Fig 1. Number of SARS-CoV-2 positive tests among individuals admitted to hospital with myocarditis or pericarditis by test week: data from entire eligible population in England.** SARS-CoV-2, Severe Acute Respiratory Syndrome Coronavirus 2.

**Table 2. Adjusted RI with 95% CIs of admissions with myocarditis in SUS using the SCCS analysis (adjusted for time period (4 weekly period)) in risk intervals after a COVID-19 vaccine or after a positive SARS-CoV-2 test stratified by age group and gender.**

| Vaccination status | | Interval (days) | Ages 16–39, n = 765 | | | | Ages 40+, n = 1,134 | | | |
|---|---|---|---|---|---|---|---|---|---|---|
| | | | Case count | Person years | RI (95% CI) | p-Value | Case count | Person years | RI (95% CI) | p-Value |
| Baseline | | | 486 | 544 | | | 823 | 801.3 | | |
| ChAdOx1-S | Dose 1 | 0 to 6 | 9 | 2.2 | 5.23 (2.48, 11.01) | <0.001 | 11 | 7.8 | 1.45 (0.78, 2.71) | 0.238 |
| | | 7 to 13 | n < 2 | 2.4 | | | 9 | 8.9 | 0.98 (0.5, 1.93) | 0.955 |
| | Dose 2 | 0 to 6 | 2 | 2.6 | 1.01 (0.25, 4.13) | 0.991 | 7 | 12.5 | 0.53 (0.25, 1.12) | 0.097 |
| | | 7 to 13 | 6 | 2.6 | 2.88 (1.25, 6.65) | 0.013 | 7 | 12.4 | 0.53 (0.25, 1.13) | 0.101 |
| BNT162b2 | Dose 1 | 0 to 6 | 18 | 9 | 2.23 (1.37, 3.63) | 0.001 | 5 | 2.3 | 2.01 (0.78, 5.2) | 0.151 |
| | | 7 to 13 | 14 | 9.2 | 1.62 (0.94, 2.79) | 0.084 | 3 | 2.8 | 1.03 (0.32, 3.3) | 0.965 |
| | Dose 2 | 0 to 6 | 40 | 8 | 5.34 (3.81, 7.48) | <0.001 | 6 | 7.2 | 0.79 (0.35, 1.8) | 0.578 |
| | | 7 to 13 | 17 | 7.9 | 2.25 (1.37, 3.68) | 0.001 | 5 | 7.2 | 0.67 (0.27, 1.62) | 0.37 |
| | Booster | 0 to 6 | 17 | 3.5 | 4.38 (2.59, 7.38) | <0.001 | 22 | 13.2 | 1.5 (0.97, 2.33) | 0.07 |
| | | 7 to 13 | 6 | 3.3 | 1.65 (0.72, 3.78) | 0.237 | 24 | 13.1 | 1.62 (1.06, 2.47) | 0.025 |
| mRNA-1273 | Dose 1 | 0 to 6 | 8 | 1.8 | 8.69 (4.01, 18.81) | <0.001 | 2 | 0.3 | 7.02 (1.53, 32.23) | 0.012 |
| | | 7 to 13 | n < 2 | 1.8 | | | n < 2 | 0.3 | | |
| | Dose 2 | 0 to 6 | 38 | 1.4 | 56.48 (33.95, 93.97) | <0.001 | n < 2 | 0.2 | | |
| | | 7 to 13 | n < 2 | 1.4 | | | n < 2 | 0.2 | | |
| | Booster | 0 to 6 | 11 | 1.4 | 7.88 (4.02, 15.44) | <0.001 | 3 | 2.8 | 0.88 (0.28, 2.8) | 0.827 |
| | | 7 to 13 | n < 2 | 1.4 | | | 5 | 2.8 | 1.49 (0.6, 3.69) | 0.39 |
| COVID infection test day 0 | | | 10 | 1.1 | 8.23 (4.34, 15.6) | <0.001 | 20 | 0.9 | 25.6 (16.09, 40.73) | <0.001 |
| Post COVID infection: 1–27d | | | 49 | 28.7 | 1.67 (1.22, 2.28) | <0.001 | 72 | 24 | 3.64 (2.76, 4.81) | <0.001 |

| Vaccination status | | Interval (days) | All 16+ year olds: males, n = 1,194 | | | | All 16+ year olds: females, n = 721 | | | |
|---|---|---|---|---|---|---|---|---|---|---|
| | | | Case count | Person years | RI (95% CI) | p-Value | Case count | Person years | RI (95% CI) | p-Value |
| Baseline | | | 792 | 844.7 | | | 523 | 513.2 | | |
| ChAdOx1-S | Dose 1 | 0 to 6 | 11 | 6.3 | 2.01 (1.08, 3.76) | 0.028 | 9 | 3.7 | 2.75 (1.35, 5.62) | 0.006 |
| | | 7 to 13 | 6 | 6.9 | 0.97 (0.43, 2.21) | 0.941 | 4 | 4.4 | 1.05 (0.38, 2.89) | 0.92 |
| | Dose 2 | 0 to 6 | 6 | 8.9 | 0.79 (0.35, 1.78) | 0.563 | 3 | 6.2 | 0.5 (0.16, 1.58) | 0.239 |
| | | 7 to 13 | 10 | 8.9 | 1.32 (0.7, 2.49) | 0.398 | 3 | 6.2 | 0.49 (0.16, 1.54) | 0.221 |
| BNT162b2 | Dose 1 | 0 to 6 | 16 | 8.5 | 2.04 (1.22, 3.42) | 0.006 | 8 | 3.2 | 2.62 (1.28, 5.36) | 0.009 |
| | | 7 to 13 | 13 | 8.9 | 1.59 (0.91, 2.78) | 0.105 | 7 | 3.4 | 1.9 (0.88, 4.1) | 0.103 |
| | Dose 2 | 0 to 6 | 40 | 9.7 | 4.42 (3.18, 6.14) | <0.001 | 8 | 5.6 | 1.6 (0.78, 3.27) | 0.196 |
| | | 7 to 13 | 16 | 9.6 | 1.78 (1.08, 2.94) | 0.025 | 6 | 5.6 | 1.18 (0.52, 2.68) | 0.685 |
| | Booster | 0 to 6 | 22 | 9.4 | 2.01 (1.29, 3.11) | 0.002 | 17 | 7.3 | 2 (1.21, 3.3) | 0.007 |
| | | 7 to 13 | 18 | 9.2 | 1.64 (1.01, 2.65) | 0.044 | 12 | 7.2 | 1.45 (0.8, 2.61) | 0.219 |
| mRNA-1273 | Dose 1 | 0 to 6 | 9 | 1.7 | 10.48 (5.02, 21.87) | <0.001 | n < 2 | 0.4 | | |
| | | 7 to 13 | n < 2 | 1.7 | | | n < 2 | 0.4 | | |
| | Dose 2 | 0 to 6 | 35 | 1.3 | 58.24 (34.78, 97.52) | <0.001 | 4 | 0.2 | 23.26 (6.68, 81.06) | <0.001 |
| | | 7 to 13 | n < 2 | 1.3 | | | n < 2 | 0.2 | | |
| | Booster | 0 to 6 | 12 | 2.7 | 3.8 (2.06, 7.01) | <0.001 | 2 | 1.5 | 1.11 (0.27, 4.61) | 0.882 |
| | | 7 to 13 | 3 | 2.7 | 0.99 (0.31, 3.15) | 0.99 | 3 | 1.4 | 1.72 (0.53, 5.56) | 0.363 |
| COVID infection test day 0 | | | 21 | 1.3 | 17.19 (10.98, 26.89) | <0.001 | 10 | 0.8 | 13.31 (6.99, 25.32) | <0.001 |
| Post COVID infection: 1–27d | | | 76 | 32.8 | 2.5 (1.93, 3.24) | <0.001 | 47 | 20.6 | 2.47 (1.77, 3.45) | <0.001 |

CI, confidence interval; COVID-19, Coronavirus Disease 2019; RI, relative incidence; SARS-CoV-2, Severe Acute Respiratory Syndrome Coronavirus 2; SCCS, self-controlled case series; SUS, Secondary Uses Service.

[14.58, 60.4], $p < 0.001$), respectively, with the elevated risk predominantly seen in males (Table A in S2 Appendix).

There was less evidence of a vaccine-associated risk for pericarditis admissions apart from 0 to 6 days after a second dose of mRNA-1273 vaccine in 16 to 39 year olds, RI 4.82 (95% CI [1.82, 14.01], $p = 0.004$ (Table B in S2 Appendix)). However, ECDS visits for both pericarditis and myocarditis showed elevated risks in 16 to 39 year olds that paralleled those seen in the myocarditis admissions (Table C in S2 Appendix).

When stratified by the vaccine given for priming, both mRNA vaccines showed evidence of an elevated risk (myocarditis and pericarditis combined) in the 0- to 6-day period when given as a booster after 2 priming doses of the BNT162b2 vaccine in 16 to 39 year olds—RI 3.23 (95% CI [1.85, 5.64], $p < 0.001$) for a BNT162b2 booster and 6.06 (95% CI [2.95, 5.14], $p < 0.001$); there was insufficient data to evaluate the risk after 3 doses of the mRNA-1273. RIs were not significantly elevated when BNT162b2 or mRNA-1273 booster doses were given after a priming course of ChAdOx1-S vaccine (Table D in S2 Appendix).

The results of the cohort analysis of hospital admissions were similar to those obtained in the SCCS analysis when stratified by age and, for the booster effect, when stratified by the vaccine given for priming (Table E in S2 Appendix). For the 40+ age group RIs from 14+ days after a booster dose were below one when an mRNA vaccine was given after ChAdOx1-S or BNT162b2 priming.

Results were similar in the sensitivity analysis in which the study period ended on 23 August 2021 prior to the MHRA advice about the risk of myocarditis with mRNA vaccines, though with generally lower RI estimates (Fig 2, Table F in S2 Appendix).

When restricted to individuals with a prior confirmed SARS-CoV-2 infection RI estimates in 16 to 39 year olds were generally lower than in those who were infection naïve though elevated RIs were still evident after a second dose of BNT162b2 and mRNA-1273 vaccines in those with prior infection—RI 2.47 (95% CI [1.32, 4.63], $p = 0.005$) and 19.07 (95% CI [8.62, 42.19], $p < 0.001$), respectively, and also after booster doses of these vaccines (Table G in S2 Appendix). For ChAdOx1-S, all the cases after a first dose were in those who were infection naïve, RI 4.21 (2.04, 8.72).

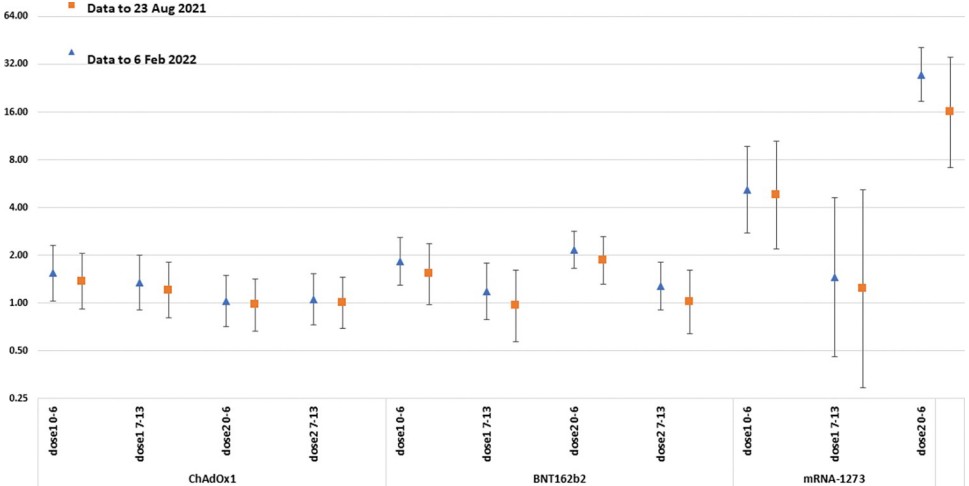

**Fig 2. Comparing the RI estimates from the SCCS analysis of myocarditis or pericarditis using hospital admissions for whole study period (6 February 2022) with the truncated 23 August 2021 period.** All ages, by vaccine type. RI, relative incidence; SCCS, self-controlled case series.

Vaccination of 12 to 15 year olds only began on 13 September 2021 and was restricted to the use of the mRNA vaccines, predominantly BNT162b2 in the study period. The results of the cohort and SCCS analysis for emergency care consultations for this age group showed an elevated risk of an ECDS attendance in the 0 to 6 day and 7 to 13 days after BNT162b2 vaccine for myocarditis or pericarditis with similar risks after the first and second doses (Table H in S2 Appendix). For hospital admissions, there were few cases in this age group (Table 1) and only the cohort analysis was conducted for which an elevated RI of 12.16 (95% CI [2.21, 66.7], $p = 0.004$), 0 to 6 days after a second dose was found.

## Postinfection analyses

An elevated RI of admission for myocarditis 1 to 27 days after a confirmed SARS-CoV-2 infection was evident (Table 2); RI estimates were higher in ≥40 than 16 to 39 year olds and higher in those tested on the day of hospital admission (day 0). The post SARS-CoV-2 infection risk was similar between genders. Pericarditis admissions also showed elevated risks within the month after a SARS-CoV-2 infection (Table B in S2 Appendix).

Stratification of the postinfection risk of myocarditis or pericarditis by first versus second or subsequent infections, by vaccination status and by variant showed no significant differences between RIs for the day 1 to 27 risk period, though point estimates were lower for breakthrough infections and for Omicron than earlier variants (Table 3).

## Attributable risk

The attributable risk estimates from the SCCS analysis for hospital admissions for myocarditis or pericarditis after vaccination and after a SARS-CoV-2 infection are shown in Table 4 for exposures with $p < 0.001$ for an elevated RI estimate. Attributable risk estimates for admission for myocarditis or pericarditis within 0 to 27 days of a laboratory-confirmed were 10.2 and 18.1 per million in 16 to 39 and ≥40 year olds, respectively.

For 12 to 15 year olds, the attributable risk for an admission 0 to 6 days after a second dose based on a total of 2 cases was 3.0 (95% CI [2.1, 3.7]) per million.

**Table 3. RI of infection by the SCCS method (adjusted for time period (4 weekly period)) for hospital admitted patients with myocarditis or pericarditis aged 12 + years who tested positive for SARS-CoV-2 infection.**

| Interval from test to admission | Type of infection | | Vaccination status | | Variant | | |
|---|---|---|---|---|---|---|---|
| | First | Reinfection | Unvaccinated^ | Vaccinated ≥1 dose | Alpha | Delta | Omicron |
| **Day 0** | | | | | | | |
| Number cases | 50 | 4 | 19 | 35 | 3 | 29 | 22 |
| Person years | 3.1 | 0.9 | 1 | 3.1 | 0.3 | 2.3 | 1.4 |
| RI (95% CIs) | 17.29 | 4.75 | 24.66 | 11.98 | 13.96 | 13.18 | 16.79 |
| | (12.96, 23.00) | (1.77, 12.80) | (15.42, 39.44) | (8.39, 16.59) | (4.37, 46.64) | (9.06, 19.17) | (10.83, 26.01) |
| | $p < 0.001$ | $p = 0.002$ | $p < 0.001$ | $p < 0.001$ | $p < 0.001$ | $p < 0.001$ | $p < 0.001$ |
| **Days 1–27** | | | | | | | |
| Number cases | 193 | 50 | 71 | 172 | 27 | 150 | 66 |
| Person years | 82.4 | 22.5 | 27 | 77.8 | 10.8 | 62.1 | 32 |
| RI (95% CIs) | 2.57 | 2.49 | 3.32 | 2.33 | 3.13 | 2.58 | 2.34 |
| | (2.19, 3.01) | (1.82, 3.41) | (2.54, 4.33) | (1.96, 2.76) | (2.02, 4.84) | (2.15, 3.1) | (1.77, 3.1) |
| | $p < 0.001$ | $p < 0.001$ | $p < 0.001$ | $p < 0.001$ | $p < 0.001$ | $p < 0.001$ | $p < 0.001$ |

^ Includes infections before the first dose.

CI, confidence interval; RI, relative incidence; SARS-CoV-2, Severe Acute Respiratory Syndrome Coronavirus 2; SCCS, self-controlled case series.

**Table 4. Attributable risk estimates with 95% CIs for an admission for myocarditis; exposures with elevated RIs with *p* < 0.001 0–6 days post-vaccine and 0 to 27 days post a laboratory confirmed SARS-CoV-2 in the SCCS analysis using the dataset that includes infections in unvaccinated individuals.**

| Vaccine exposure<br>0 to 6 days before admission 16 to 39 year olds | Admissions in risk period | RI | Attributable fraction | Attributable cases | Doses | Attributable risk per million vaccinations (95% CI) |
|---|---|---|---|---|---|---|
| Dose 1 ChAdOx1 | 9 | 5.23 | 0.809 | 7.3 | 2,099,584 | 3.5 (2.6,3.9) |
| Dose 1 BNT162b2 | 18 | 2.23 | 0.552 | 9.9 | 9,705,976 | 1.02 (0.5,1.3) |
| Dose 2 BNT162b2 | 40 | 5.34 | 0.813 | 32.5 | 9,519,878 | 3.4 (3.1,3.6) |
| Booster BNT 162b2 | 17 | 4.38 | 0.772 | 13.1 | 5,319,875 | 2.5 (2.0,2.8) |
| Dose 1 mRNA 1273 | 8 | 8.69 | 0.885 | 7.1 | 1,023,825 | 6.9 (5.9,7.4) |
| Dose 2 mRNA 1273 | 38 | 56.48 | 0.982 | 37.3 | 881,058 | 42.4 (41.9,42.7) |
| Booster mRNA 1273 | 11 | 7.88 | 0.873 | 9.6 | 2,144,219 | 4.5 (3.9,4.8) |
| Vaccine exposure<br>0 to 6 days before admission all 16 + year olds | Admissions in risk period | RI | Attributable fraction | Attributable cases | Doses | Attributable risk per million vaccinations (95% CI) |
| Dose 2 BNT162b2: males | 40 | 4.42 | 0.774 | 31.0 | 8,734,711 | 3.5 (3.1,3.8) |
| Dose 1 mRNA 1273: males | 9 | 10.48 | 0.905 | 8.1 | 747,072 | 10.9 (9.6,11.5) |
| Dose 2 mRNA 1273: males | 35 | 58.24 | 0.983 | 34.4 | 625,871 | 55.0 (54.3,55.3) |
| Booster mRNA 1273: males | 12 | 3.8 | 0.737 | 8.8 | 3,539,697 | 2.5 (1.7,2.9) |
| Dose 2 mRNA 1273: females | 4 | 23.26 | 0.957 | 3.8 | 498,121 | 7.7 (6.8,7.9) |
| SARS-CoV-2 infection 0–27 days before admission | Admissions in risk period | RI | Attributable fraction | Attributable cases | Cases | Attributable risk per million infections |
| 16 to 39 year olds day 0 | 15 | 6.98 | 0.857 | 12.9 | 8,398,257 | 1.5 (1.4–1.6) |
| ≥40 year olds day 0 | 37 | 23.56 | 0.958 | 35.4 | 6,570,699 | 5.4 (5.6, 5.5) |
| 16–39 year olds days 1–27 | 113 | 2.09 | 0.522 | 58.9 | 8,398,257 | 7.0 (5.5, 8.3) |
| ≥40 year olds days 1–27 | 124 | 3.07 | 0.674 | 83.6 | 6,570,699 | 12.7 (11.3, 13.9) |

CI, confidence interval; RI, relative incidence; SARS-CoV-2, Severe Acute Respiratory Syndrome Coronavirus 2; SCCS, self-controlled case series.

## Discussion

Our study showed an increased risk of hospital admission with myocarditis 0 to 6 days after an mRNA vaccine, predominantly in males aged 16 to 39 years with the highest risk in those aged 16 to 24 years. An elevated risk was evident after each of the priming doses with higher risks after a second dose of the mRNA-1273 than the BNT162b2 vaccine. There was also evidence of an increased risk after a booster dose of each of the mRNA vaccines when given after a priming course of BNT162b2 but not after a primary course of ChAdOx1-S vaccine. The only elevated risk seen after the ChAdOx1-S vaccine was post-first dose in 16 to 39 year olds. An elevated risk of hospital admission after a second or booster dose of the mRNA vaccines was still present in individuals vaccinated after a confirmed SARS-CoV-2 infection, though RIs were lower than in those without a prior infection. Our study also showed an increased RI of myocarditis and pericarditis after a SARS-CoV-2 infection which was not abrogated by prior vaccination or previous infection and was evident after each infecting variant.

While there was little evidence of an elevated risk of hospital admission for pericarditis after primary or booster vaccination, emergency care consultations showed elevated risks for pericarditis in 16 to 39 year olds 0 to 6 days after a first dose of ChAdOx1-S and after primary and booster doses of the mRNA vaccines. Hospital admission for pericarditis, particularly a first attack in an otherwise healthy young adult, is usually not indicated [17]. This is reflected in Table 1 that shows 3,401 emergency care consultations in <40 year olds for pericarditis but only 561 admissions. As a consequence, many of the postvaccination risk periods in the

hospital admissions analysis for pericarditis had less than 2 cases for which RIs were not estimated. Emergency care consultations for pericarditis and myocarditis showed a similar pattern of elevated risks post-vaccine as admissions for myocarditis in adults aged 16 to 39 years which is supportive of the specificity of the diagnoses in the ECDS dataset.

Our results confirm the findings of earlier studies of the risk of myocarditis or pericarditis following primary immunisation with mRNA vaccines [4–8,12] and in addition show that while there is an elevated risk when mRNA vaccines are given as a booster dose, this is lower than after the second dose as suggested by other recent studies [13,18–20]. The difference in risk between the second and booster dose was particularly marked for the mRNA-1273 vaccine. The quantity of mRNA in mRNA-1273 when used for boosting is 50 micrograms, half the amount when used for priming [21], compared with 30 micrograms in BNT162b2 vaccine when used for priming or boosting. This suggests that the risk of myocarditis after mRNA vaccines may in part be determined by mRNA dose, possibly the amount of double-stranded RNA (dsRNA) that can generate a dose-related innate immune activation and can be present in low quantities in the mRNA vaccines [22]. The rapid onset after vaccination, even after a first dose, would be consistent with such a direct effect. Based on case reports suggesting an increased risk in those vaccinated after infection, together with the increased risk after the second dose, an immune-mediated mechanism involving antibodies to some component of the spike protein has been proposed [23–25]. However, our study showed that the vaccine risk was not exacerbated in those who had a prior SARS-CoV-2 infection, nor was the risk increased after a booster dose—scenarios which result in high levels of anti-spike IgG antibodies, particularly in younger adults [26].

The risk after ChAdOx1-S was confined to the first dose and may therefore involve a different mechanism to that induced by mRNA vaccines. Other systemic effects such as fever, headache, and malaise are also more common after a first than second dose of this vaccine [27], unlike the 2 mRNA vaccines for which systemic symptoms are more common after a second dose [28,29]. The complication of thrombosis with thrombocytopenia after ChAdOx1-S vaccine also occurs predominantly after a first dose but has a longer onset after vaccination than myocarditis suggesting an unrelated causal mechanism [16].

There was little power for assessment of the vaccine-associated risk in 12 to 15 year olds as the background rate of hospital admissions in this age group for myocarditis and pericarditis was low, around 1.1 per 100,000 person years with only 10 admissions at any time after a first or second dose of BNT162b2 vaccine. Nevertheless, the estimated attributable risk 0 to 6 days after a second dose (3 per million) was similar to that in 16 to 39 year olds (3.5 per million).

While we did not assess severity in vaccine-associated compared with unvaccinated cases, there were 20 deaths during admission or within a week of discharge among the 906 unvaccinated cases with myocarditis or pericarditis (2.2%) compared with 3 deaths during admission or within a week of discharge among the 265 cases (1.1%) with onset within 0 to 6 days of a dose of vaccine all with a diagnosis of myocarditis. Of these, 2 were after a first dose of ChAdOx1-S vaccine and a third after a booster dose of BNT162b2 given after ChAdOx1-S priming; all 3 individuals had underlying comorbidities and in addition 2 were aged 80 years or over, consistent with the early use of the ChAdOx1-S vaccine in high risk and elderly groups. The absence of fatal cases among healthy young adults with vaccine-associated myocarditis or pericarditis is reassuring and consistent with case reports of a benign outcome [14].

The post-vaccine attributable risk estimates for myocarditis or pericarditis were lower than reported in studies in some other settings. For example, in Israel, an excess risk of myocarditis in males (all ages) of 1 in 26,000 second doses of BNT162b2 vaccine (38.5 per million) was estimated. However, in Israel, clinicians were alerted to the risk of vaccine-associated myocarditis and were requested to actively report cases and, unlike England, almost all cases are admitted

to hospital in Israel [3]. Differences in attributable risk are not just dependent on relative risks in the postvaccination period but also on the background rate of ascertainment of cases of myocarditis and pericarditis, which will reflect diagnostic and admission practices in the study population. It is difficult to directly compare RI estimates across studies due to differences in the postvaccination risk periods and age/sex stratification used in the analysis, but the same RI estimates will generate higher attributable risk estimates for populations with a higher background incidence of the outcome.

Unlike the vaccine-associated risk, the risk of myocarditis or pericarditis after a confirmed SARS-CoV-2 infection was greater in older adults than those under 40 years of age and was similar for each cardiac outcome. Our RI estimates are lower than reported in earlier Israeli and English studies [4,12], both of which covered periods when the predominant variant was Alpha and prior to the emergence of Omicron which is associated with an increased risk of reinfection and breakthrough infections in vaccinated individuals [30]. The English study also restricted the analysis to first recorded SARS-CoV-2 infection. We therefore conducted an ad hoc analysis to assess the effect of variant and of prior infection or vaccination on the risk of myocarditis/pericarditis following SARS-CoV-2 infection. Although not significantly different from each other, the direction of the differences in RIs is consistent with a higher risk in the earlier period when a greater proportion of SARS-CoV-2 cases were primary infections in unvaccinated individuals before the emergence of Omicron.

Day 0 was not included in the main postinfection risk period risk because case ascertainment will be enhanced compared to other risk periods by the practice of testing patients for SARS-CoV-2 infection on admission as part of infection control procedures. In addition, while some of the day 0 cases will be causally related to infection, in others, infection may be coincidental rather than causal and only detected because of the testing policy. The high RIs observed in our study on day 0 are therefore expected and are consistent with findings in other studies of hospital admission for COVID-19 complications [31]. The RI for admission in the 21 days before a positive test was generally around 1 and not elevated as found in an SCCS analysis of thrombotic conditions associated with SARS-CoV-2 infection in Sweden that was attributed to testing delays or nosocomial infection [31].

Ours is an observational study and so has inherent limitations. There is the potential for enhanced ascertainment of cases occurring shortly after vaccination due to vaccinees' knowledge of the warnings issued by the MHRA about the risk of myocarditis with mRNA vaccines. While the postvaccination RI estimates were generally lower when the analysis was restricted to admissions before 23 August 2021 when the MHRA warning was first issued, the overall pattern of results was similar (Fig 2). However, it was not possible to make this comparison for 12 to 15 year olds, as vaccination for this age group was only routinely recommended after this date and it is possible that biased ascertainment contributed to the elevated RIs found in the age group. Another limitation is that ours was a database study without case note review so outcomes were reliant on routinely coded diagnoses in patients' notes. The effect of this is difficult to assess as outcome misclassification usually results in an underestimate of the true RI but if it differs by vaccination status (for example, a greater tendency to assign a myocarditis code in unproven vaccinated cases once the adverse event was publicised by the MHRA), then the RI would be overestimated. The RIs for myocarditis from the ECDS analysis were generally lower than for the hospital admitted cases which would suggest that despite the higher incidence of outcome events, capture of true myocarditis or pericarditis cases is less accurate. It is also possible that by restricting our myocarditis and pericarditis codes to those occurring in the first 3 diagnosis fields, we may have missed some SARS-CoV-2–associated cases who were admitted for respiratory or other non-cardiac reasons which may have led to an underestimate of the frequency with which myocarditis or pericarditis occurs in COVID-19 patients. Also,

the attributable risk estimates for COVID-19 used laboratory confirmed cases as the denominator and will be affected by the proportion of all SARS-CoV-2 infections captured by testing, precluding a direct comparison with vaccine-associated attributable risks.

While the SCCS method we used for the main analysis adjusts for potential time-invariant confounders such as comorbidities more reliably than the cohort method, the latter is better able to adjust for event-dependent exposures such as contraindication of further doses on the occurrence of an event after an earlier dose. However, as shown in S1 Appendix, there was little evidence of such a bias in our dataset. The similarity in results between the SCCS and cohort analysis suggests that the latter adequately adjusted for confounders. The lower RIs in the cohort analysis from 14 days postvaccination in those aged 40 years and older may reflect a true vaccine effect through prevention of COVID-19 and its associated risk of myocarditis which would not be detectable using the SCCS method.

In conclusion, our study provides estimates of the excess risk of an episode of a hospital admitted episode of myocarditis after a third dose of an mRNA vaccine in England which in 16 to 39 year olds was 2.5 and 4.5 per million for the BNT162b2 and mRNA-1273 vaccines, respectively. The comparable risk estimates after a second dose were 3.4 and 42.4 per million, respectively, with higher risks in males. The absence of acute fatal outcomes in healthy adults with vaccine-associated myocarditis in our study, and the lower risk of death or cardiac failure within 3 months compared with other causes of myocarditis demonstrated by others, is reassuring [32].

## Supporting information

**S1 STROBE Checklist. STROBE Checklist.**
(XLSX)

**S1 Protocol. Analysis of hospital admissions and emergency care consultations for acute myocarditis and pericarditis after COVID-19 vaccines in England.**
(PDF)

**S1 Appendix. Supplementary material.** Checking the self-controlled case series assumption of no long-term event-dependence. Figure A. Plots of preexposure and postexposure relative incidence, by preexposure period length, SUS data to 6 February 2022. Figure B. Plots of preexposure and postexposure relative incidence, by preexposure period length, ECDS data to 6 February 2022. Table A. Relative incidence estimates from the standard SCCS model and the event-dependent exposures SCCS model, SUS first cases in individuals with no recorded positive SARS-CoV-2 test before the end of the observation period (6 February 2022), $N$ = 1,977. Table B. Relative incidence estimates from the standard SCCS model and the event-dependent exposures SCCS model, ECDS first cases in individuals with no recorded positive SARS-CoV-2 test before the end of the observation period (6 February 2022), $N$ = 3,553.
(DOCX)

**S2 Appendix. Supplementary tables.** Table A. Adjusted (adjusted for time period (4 weekly period)) RI of admissions with myocarditis after a COVID-19 vaccine by postvaccination risk interval in 16–24 and 25–39 year olds and 16–39 year olds by gender in SUS using the SCCS analysis—whole study period to 6 February 2022. Table B. Adjusted (for time period (4 weekly period)) RI of admissions with pericarditis after a COVID-19 vaccine by postvaccination risk interval by age group in SUS using the SCCS analysis—whole study period to 6 February 2022. Table C. Adjusted (for time period (4 weekly period)) RI of attendances with myocarditis or pericarditis after a COVID-19 vaccine by postvaccination risk interval by age group in ECDS using the SCCS analysis—whole study period to 6 February 22. Table D. Adjusted (for time

period (4 weekly period)) RI of admissions with myocarditis or pericarditis in SUS using the SCCS analysis in risk periods after a COVID-19 vaccine or a SARS-CoV-2 infection with booster doses stratified by vaccine given for priming. Table E. Adjusted (for time period (4 weekly period)) relative risk (aRR) of attendances with myocarditis or pericarditis in SUS using a cohort analysis after a COVID-19 vaccine by postvaccination risk interval. Adjusted for time period, age group, gender, region, ethnic group, CEV, and other clinical risk group. Table F. Adjusted (for time period (4 weekly period)) RI of admissions with myocarditis or pericarditis after a COVID-19 vaccine by postvaccination risk interval in SUS using the SCCS analysis with data up to 23 August 2021. Table G. Adjusted (for time period (4 weekly period)) RI of hospital admission in SUS with myocarditis or pericarditis after a COVID-19 vaccine by postvaccination risk interval in 16–39 year olds with or without a prior SARS-CoV-2 infection using the SCCS analysis—whole study period to 6 February 22. Table H. Relative incidence (adjusted for time period (4 weekly period)) using the SCCS analysis and aRR of attendances in 12–15 year olds with myocarditis or pericarditis in ECDS after a BNT162b2 COVID-19 vaccine by postvaccination risk interval.
(DOCX)

## Author Contributions

**Conceptualization:** Julia Stowe, Elizabeth Miller, Nick Andrews, Heather J. Whitaker.

**Data curation:** Julia Stowe.

**Formal analysis:** Nick Andrews, Heather J. Whitaker.

**Investigation:** Elizabeth Miller.

**Methodology:** Julia Stowe, Elizabeth Miller, Heather J. Whitaker.

**Writing – original draft:** Elizabeth Miller.

**Writing – review & editing:** Julia Stowe, Elizabeth Miller, Nick Andrews, Heather J. Whitaker.

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
