## [Editor Report · Decision Letter 0]

21 Sep 2022

Dear Dr Miller, 

Thank you for submitting your manuscript entitled "Risk of myocarditis after a COVID-19 mRNA vaccine booster following homologous or heterologous priming in those with and without prior SARS-CoV-2 infection; an observational database study in England" for consideration by PLOS Medicine.

Your manuscript has now been evaluated by the PLOS Medicine editorial staff and I am writing to let you know that we would like to send your submission out for external peer review.

Please re-submit your manuscript within two working days, i.e. by Sep 23 2022 11:59PM.

Kind regards,

Philippa Dodd, MBBS MRCP PhD

PLOS Medicine

---

## [Decision Letter · Decision Letter 1]

2 Feb 2023

Dear Dr. Miller,

Thank you very much for submitting your manuscript "Risk of myocarditis after a COVID-19 mRNA vaccine booster following homologous or heterologous priming in those with and without prior SARS-CoV-2 infection; an observational database study in England" (PMEDICINE-D-22-03092R1) for consideration at PLOS Medicine. 

Your paper was evaluated by a senior editor and discussed among all the editors here. It was also sent to independent reviewers, including a statistical reviewer. The reviews are appended at the bottom of this email and any accompanying reviewer attachments can be seen via the link below:

[LINK]

In light of these reviews, I am afraid that we will not be able to accept the manuscript for publication in the journal in its current form, but we would like to consider a revised version that addresses the reviewers' and editors' comments. Obviously we cannot make any decision about publication until we have seen the revised manuscript and your response, and we plan to seek re-review by one or more of the reviewers. 

We expect to receive your revised manuscript by Feb 23 2023 11:59PM. Please email us (plosmedicine@plos.org) if you have any questions or concerns.

We look forward to receiving your revised manuscript. 

Sincerely,

Philippa Dodd, MBBS MRCP PhD

PLOS Medicine

plosmedicine.org

GENERAL

Please respond to all editor and reviewer comments detailed below, in full.

DATA AVAILABILITY STATEMENT

PLOS Medicine requires that the de-identified data underlying the specific results in a published article be made available, without restrictions on access, in a public repository or as Supporting Information at the time of article publication, provided it is legal and ethical to do so. Please see the policy at 

http://journals.plos.org/plosmedicine/s/data-availability

and FAQs at 

http://journals.plos.org/plosmedicine/s/data-availability#loc-faqs-for-data-policy

PLOS defines the “minimal data set” to consist of the data set used to reach the conclusions drawn in the manuscript with related metadata and methods, and any additional data required to replicate the reported study findings in their entirety. Authors do not need to submit their entire data set, or the raw data collected during an investigation. Please submit the following data:

The values behind the means, standard deviations and other measures reported;

The values used to build graphs;

The points extracted from images for analysis.

The Data Availability Statement (DAS) requires revision. For each data source used in your study: 

TITLE

Suggest “Risk of myocarditis after a COVID-19 mRNA vaccine booster following homologous or heterologous priming in those with and without prior SARS-CoV-2 infection: a self-controlled case series analysis in England” or something similar. You may wish to make further revisions in accordance with reviewer comments – please see below.

ABSTRACT

Abstract Background: 

It may be helpful to expand this section further outlining why the study is important. Please ensure that the final sentence clearly states the study question.

Abstract methods and findings: 

Line 33-34: is the word “days” missing after 1-27?

Please include the timeframe over which the data were collected and the main outcome measures.

Please include the number of participants (of the 50 million eligible to receive the vaccine) that were diagnosed with myocarditis and/or pericarditis (7929 individuals as per the results section of the main manuscript)

Please detail where data were extracted from (i.e. which databases)

Please quantify the main results with p values as well as 95% CIs. When reporting p values please report as p<0.001 (not .001) or where higher, the exact p value as p=0.002 (not .002), for example. Suggest modifying statistical reporting to read as follows: 3.23 (95% CI [1.85,5.64]; p= 0.0… or p<0.001…) and 6.06 (95% CI [2.95,12.47], p=0.00 or p<0.001)

Please include any important dependent variables that are adjusted for in the analyses.

Apart from the possibility of overestimation of the RIs, were there any other limitations of the study's methodology? 

Abstract Conclusions: 

Line 41: Please address the study implications without overreaching what can be concluded from the data; the phrase "In this study, we observed ..." may be useful. This sentence is rather long and as result not very accessible to the reader, please revise.

Line 43: suggest “…which contains less mRNA...” perhaps

Line 46: Suggest “…behind this phenomenon…” perhaps

Please interpret the study based on the results presented in the abstract, emphasizing what is new without overstating your conclusions.

AUTHOR SUMMARY

Thank you for including an author summary. Please ensure that each bullet point is succinct, and that the language used is accessible to the general reader. It may be helpful for you to visit the journal's homepage here: https://journals.plos.org/plosmedicine/ for published examples.

Why was this study done?

Line 49 Please add the heading “Author Summary” before the sub-headings

Please ensure under each sub-heading, individual statements follow bullet-points.

What did the researchers do and find?

This section is rather long. Please trim to no more than 4 individual bullet points

Line 80: please revise the sub-heading to read “What do these findings mean?”

INTRODUCTION

Lines 101 and 102: please only include non-proprietary names for vaccines

Line 119: Sentence beginning “Given…” suggest moving to an appropriate part of the methods section 

METHODS and RESULTS

Lines 192: see statistical reviewer comments also, which we agree with, please provide brief details of your statistical methods here. For all observational studies, we ask that in the manuscript text you indicate: 

(1) the specific hypotheses you intended to test, 

(2) the analytical methods by which you planned to test them, 

(3) the analyses you actually performed, and 

(4) when reported analyses differ from those that were planned, transparent explanations for differences that affect the reliability of the study's results. If a reported analysis was performed based on an interesting but unanticipated pattern in the data, please be clear that the analysis was data-driven.

Lines 124: please include the number of residents (50 million as per the abstract)

Line 125: please define “outcomes of interest”

Line 159 onwards: please define PCR, LFT, UKHSA at first use

Line 166: please define NHS at first use

Please remove governance, funding, data availability and conflicts of interest statements and include only in the relevant sections of the manuscript submission form. In the event of publication they will be complied as metadata

As for the abstract, please quantify the main results with p values as well as 95% CIs. When reporting p values please report as p<0.001 (not .001) or where higher, the exact p value as p=0.002 (not .002), for example. Suggest modifying statistical reporting to read as follows: 3.23 (95% CI [1.85,5.64]; p= 0.0… or p<0.001…) and 6.06 (95% CI [2.95,12.47], p=0.00 or p<0.001). Please detail the statistical test (s) used to drive p values.

FIGURES 

Please define all abbreviations including those used in the titles/captions in an appropriate footnote

Figure 1: in the caption please define “wk” 

Figure 2: in an appropriate footnote, please indicate the meaning of the dots and lines

Figure 3: please include the year in the legend depicting the August date

TABLES

Please also see statistical reviewer comments which we agree with

As above, please define all abbreviations including those used in statistical reporting such as CI, for example

To help facilitate transparent data reporting where adjusted analyses are presented, please also include unadjusted analyses for comparison. In a caption/footnote please state which variables are adjusted for.

Throughout where relevant please include a column for reporting p values instead of asterisks. 

Please report p values as p <0.001 and where higher the exact p value. In an appropriate footnote, please detail the tests used to determine p values

Table 1: please see statistical reviewer comments (to include 95% CIs) which we agree with

Table 3: please use commas to separate upper and lower confidence limits instead of hyphens as these can be confused with the presentation of negative values, and to ensure consistency in statistical reporting. Please check and amend where necessary including supplementary files

SUPPLEMENTARY FILES

Throughout where relevant please include a column in the tables for reporting p values (instead of asterisks). Please indicate in a footnote the statistical test(s) used to determine them 

Please ensure all abbreviations used for statistical reporting are appropriately defined including CI

Figure S3A – suggest full terms instead of myo- or peri-

REFERENCES

For in-text reference callouts please ensure citations are separated by commas without spaces. For example, line 364, “…proposed [21, 22, 23].” Should read “…proposed [21,22,23].”

Comments from the reviewers:

Reviewer #2: Thank you for the opportunity to review this manuscript. It was easy to read and the analysis appears to be appropriate. The question of myocarditis risk after booster doses is important. Reassuring to see that while there still is an effect, it appears to be attenuated. The addition of infection including breakthrough infections and re-infections adds valuable perspective to our current situation. 

1) The main issue is the overlap with the Patone paper1, which is also an SCCS analysis of booster vaccination/infection in the same data. The Patone paper has follow-up among 13+ years of age from December 1, 2020, to December 15, 2021 compared to the this study which has follow-up among 12+ years of age from February 22, 2021 to February 6, 2022. But otherwise, it appears to be very similar with respect to data and analysis. It needs to be clarified in the manuscript what the key differences are and how this adds significantly to the Patone analysis.

2) Another main issue is the combination of myocarditis and pericarditis into one outcome. Myocarditis and pericarditis background rates differ by age and sex, which means that it is difficult to advice public health policy from numbers based on a combined outcome. I suggest that they are analysed separately throughout - myocarditis results could be presented in text and pericarditis in supplementary.

3) Abstract could be shortened (and made more readable) by reducing the number of results presented here - identify and focus on the key findings.

4) The introduction/discussion is outdated. Should be updated to include the newest studies on booster vaccination and myocarditis risk. (1-3) 

5) I would not include emergency care outcomes at all. I expect them to have poor validity. I would restrict the main analysis to hospital in-patients. Maybe place the emergency care outcomes in supplementary only?

6) Lines 189-190: Please elaborate on why follow-up was not censored at event.

7) The figures are difficult to review due to poor quality.

8) The attributable risk results should ideally also be presented with 95% CIs to reflect the statistical precision of the estimates.

9) Can you comment on the impact of the testing policy / capture of infections in England? E.g. attributable risks will be dependent on background rates of infections. How many infections do you estimate that you have captured by tests in England?

1. Patone M, Mei XW, Handunnetthi L, et al. Risk of Myocarditis After Sequential Doses of COVID-19 Vaccine and SARS-CoV-2 Infection by Age and Sex. Circulation. 2022;146:743-754. doi:10.1161/circulationaha.122.059970

2. Naveed Z, Li J, Spencer M, et al. Observed versus expected rates of myocarditis after SARS-CoV-2 vaccination: a population-based cohort study. CMAJ. 2022;194(45):E1529-E1536. doi:10.1503/CMAJ.220676

3. Stéphane Le Vu A, Bertrand M, Jabagi MJ, et al. Risk of Myocarditis after Covid-19 mRNA Vaccination: Impact of Booster Dose and Dosing Interval. medRxiv. Published online August 1, 2022:2022.07.31.22278064. doi:10.1101/2022.07.31.22278064

Reviewer #3: 

This is a very well written paper on an important topic. Also, the statistical analyses used are adequate. However, there are some concerns. My main concern is the combination of myocarditis and pericarditis combined. Apart from, maybe, diluting, the effect of vaccination on risk of vaccination in young males it also raised the question whether the data sources actually capture all cases? 

The Title is misleading as the presented comparison is actually on myocarditis and pericarditis combined. 

Page 9, 197 exposure and short-term event independency. You did assess this by inclusion of a pre-exposure period. Would be nice to just see a plot of number of days since myocarditis patients were vaccinated? I would guess very few cases would actually be vaccinated within 1-3 weeks after an event? 

Page 10, 209. A bit strange and uncommon in epidemiology to define "strong" evidence based on the p-value. :-) 

Results

Page 12, 253/ Table 1. Myocarditis and pericarditis are different diseases and their incidence vary by age. Myocarditis is more common in younger ages and pericarditis in older. Most studies on myo/peri risk after vaccination have done separate analyses on these two conditions. Why was that not done here? Presentation of incidence rates by age as in table1 for these conditions combined is not informative. Suggest to present incidence by age and sex for myo and peri separately (and peri can be in supplement). And subsequently all analyses separately for myo/peri ( I.e., suggest S3C and D as main tables). 

Also, is there any information on the validity of these diagnoses in the registers used? This in regards to the (much) lower incidence found in your data compared to others.

Also, the risk has been shown to be highest in 16-24 (rather than in 25-39). Would suggest to also present this age group, (16-24, 25-39). 

Death. Is it only possible to assess death within the first week (0-6 days) and not for eg after 28 days? Also, presentation of deaths (as all results!) should be separately for myo and peri. Do you not find any death in young (below 40) after "vaccine-induced" myocarditis in your data? 

Page 17, 387 paragraph comparing with other studies. Would suggest to compare effect estimates for the most important groups ( i.e., myocarditis in young men). As stated now you appear to find one fourth of the overall incidence of myo and peri as found in the Nordic study. To better understand the validity of your data I would suggest presenting numbers for young men and myocarditis only (not combined with pericarditis). 

Conclusion.

Page19, 449. Would avoid mentioning the comparison of outcome after vaccination with outcome after a positive covid test as the latter, as you also mention, is (highly) dependent on testing strategy (eg. When admitted to hospital). The fewer less severe tested the "higher" the risk of outcome after covid. The benefit-risk analyses is not based on comparing risk of myo after vaccination with that after covid. Hence, would avoid supporting such comparisons. 

Tables. Would suggest removing the "two-star" "three-star" significance throughout :-) 

Table 5. I guess, column heading should be " attributable… per million vaccinations" for the data regarding the vaccinees. 

Another recent reference that might be of interest:

Booster Vaccination with SARS-CoV-2 mRNA Vaccines and Myocarditis Risk in Adolescents and Young Adults: A Nordic Cohort Study of 8.9 Million Residents | medRxiv

[LINK]

---

## [Decision Letter · Decision Letter 2]

5 Apr 2023

Dear Dr. Miller,

Thank you very much for re-submitting your manuscript "Risk of myocarditis after a COVID-19 mRNA vaccine booster and after COVID-19 in those with and without prior SARS-CoV-2 infection; a self-controlled case series analysis in England" (PMEDICINE-D-22-03092R2) for review by PLOS Medicine.

I have discussed the paper with my colleagues and the academic editor and it was also seen again by 3 reviewers. I am pleased to say that provided the remaining editorial and production issues are dealt with we are planning to accept the paper for publication in the journal.

[LINK]

We look forward to receiving the revised manuscript by Apr 12 2023 11:59PM.   

Sincerely,

Philippa Dodd, MBBS MRCP PhD

PLOS Medicine

plosmedicine.org

Requests from Editors:

GENERAL

Thank you for your very detailed and considered responses to previous editor and reviewer comments, please see below for further comments which we require you address prior to publication.

Please ensure that the study is reported according to the STROBE guideline, and include the completed STROBE checklist as Supporting Information. Please add the following statement, or similar, to the Methods: "This study is reported as per the Strengthening the Reporting of Observational Studies in Epidemiology (STROBE) guideline (S1 Checklist)."

When completing the checklist, please use section and paragraph numbers, rather than page or line numbers as these often change at publication.

*** Please see statistical reviewer comments regarding the primary objectives and analyses which we agree with and require that you address in full ***

DATA AVAILABILITY STATEMENT

Thank you for updating your statement as follows:

“The raw study data are protected and are not freely available due to data privacy laws. This work is carried out under Regulation 3 of The Health Service (Control of Patient Information) (Secretary of State for Health, 2002))(3) using patient identification information without individual patient consent. Data cannot be made publicly available for ethical and legal reasons, i.e. public availability would compromise patient confidentiality as data tables list single counts of individuals rather than aggregated data. Requests for the underlying data should be made via the UKHSA office for data release: https://www.gov.uk/government/publications/accessingukhsa-protected-data”

Please include this updated statement in the manuscript submission form when you re-submit your manuscript.

TITLE

Thank you for revising your title. We suggest that the title should read as follows:

“Risk of myocarditis and pericarditis after a COVID-19 mRNA vaccine booster and after COVID-19 in those with and without prior SARS-CoV-2 infection: A self-controlled case series analysis in England”

we understand that (most) data pertaining to pericarditis has been transferred to the appendix, but pericarditis is mentioned throughout the abstract, author summary and included in the main analyses. See comments below re: abstract.

ABSTRACT

Pericarditis is not mentioned in the background section (or the title) but is throughout the rest of the abstract. Suggest introducing it in the “Background” section and (as above) including “pericarditis” in the title

Line 45: “…[22.18,62.38];p=<0.001…” should it be = or < please amend as necessary

Pericarditis is also not mentioned in the conclusions. Should it be?

Line 53: “…half the amount of mRNA…” is it exactly half? If not then perhaps “…substantially less…” or similar instead

AUTHOR SUMMARY

Thank you for making revisions to the author summary. 

In your discussion you state the following “While there was little evidence of an elevated risk of hospital admission for pericarditis after primary or booster vaccination, emergency care consultations showed elevated risks for pericarditis in 16-39 year olds 0-6 days after a first dose of ChAdOx1-S and after primary and booster doses of the mRNA vaccines” However, your summary implicates mRNA vaccines only which could be misleading, please revise.

Finally, line 92 (2nd bullet point what do these findings mean) – can this statement be made based on these data/this study design? We suggest that you consider removing this statement as it might be a bit of an over statement. Please see below for further suggested revisions.

Why the study was done?

* Primary and booster immunisation with mRNA COVID-19 vaccine have been associated with an increased risk of acute myocarditis.

* SARS-CoV-2 infection may itself cause myocarditis or pericarditis. 

* However, the effect of prior vaccination on this risk, and on the risk after a reinfection, has not been investigated.

What did the researchers do and find?

* We conducted a nationwide study in England to assess the risk of hospital admission for myocarditis or pericarditis after primary or booster vaccination, and the risk after a confirmed SARS-CoV-2 infection in those with and without confirmed previous infection.

* Elevated risks of myocarditis were found up to 6 days after each priming dose of the available mRNA vaccines (BNT162b2 and mRNA-1723) and after mRNA booster doses following a mRNA priming course but not after a priming course of the adenovirus-vectored vaccine ChAdOx1-S.

* For both mRNA vaccines, elevated risks were found in those under 40 years old, predominantly in males, were highest after the second priming dose and were generally lower in those vaccinated after a prior SARS-CoV-2 infection.

*There was an elevated risk of myocarditis and pericarditis in the 27 days after a SARS-CoV-2 infection which was higher in ≥ 40 year olds than 16-39 year olds and was still present in those with a re-infection or who had been vaccinated before infection

What do these findings mean?

* This study provides information for policy makers and those recommended to receive booster mRNA vaccines on the associated risk of myocarditis or pericarditis in a population with a high prevalence of prior SARS-CoV-2 infection.

* The greater risk associated with mRNA-1273 vaccines, which have a higher mRNA dose than BNT162b2 vaccines and, the substantially lower risk after the mRNA-booster which has half the mRNA content than used for priming, may be suggestive of a mRNA dose related mechanism but further work is required to determine this.

METHODS

Line 142: “…for the myocarditis…” please revise for improved grammar

Line 194: please revise to “Construction of the Self-Controlled Case-Series dataset (SCCS)”

TABLES

Table 1: numerical values are not well aligned to the column headers, please revise to improve clarity

Table 2: 

Title “: Adjusted*…” what does this single asterisk denote?

Thank you for including p values. You responded, “we retained asterisks to aid the reader identify outcomes that meet our pre-specified p values and which we draw attention to in the text”

Please remove the asterisks used to denote p-values. These are redundant in view of the column of p values and there is no mention of pre-specified values in the table caption so it doesn’t really help the reader. All information pertaining to the contents of the tables (and figures) should be detailed in a caption without the need to refer to the text.

To improve accessibility to the reader, please ensure upper and lower bounds of 95% CIs are on a single line not split across two lines

Table 3: throughout, please use lowercase p. please add “p=” to “reinfection day 0” 

Table 4: 

“elevated RIs (p<0.001)…” what does this p value refer to in the table?

Please define numerical values contained within parentheses (end column – attributable risk per million vaccinations)

Table S3A: as above, please remove asterisks and please ensure upper and lower bounds of 95% CIs are on a single line

SOCIAL MEDIA

To help us extend the reach of your research, please provide any Twitter handle(s) that would be appropriate to tag, including your own, your coauthors’, your institution, funder, or lab. Please detail any handles you wish to be included when we tweet this paper, in the manuscript submission form when you re-submit the manuscript.

Comments from Reviewers:

Reviewer #2: Thank you for taking my comments carefully into account. Good luck with the paper!

Reviewer #3: Table 1: please see statistical reviewer comments (to include 95% CIs) which we agree with Author response: We have added CIs to all tables apart from table 1. Since the data in table 1 is from the whole population of England and are not estimates from a sample of the population it is not appropriate to show CIs. We have pointed this out to other journals in which data from the entire population in England has been shown and for which a request for RIs was made and it has been accepted (see for example table 1 Characteristics of Persons Tested for SARS-CoV-2 in England, According to Test Positivity or Negativity and Variant in Covid-19 Vaccine Effectiveness against the Omicron (B.1.1.529) Variant | NEJM

[Minor comment: I agree with the authors. But, sometimes CI can be motivated by seeing national data as a sample in time ( i.e. it is not a sample of the population, but the population differs by years, hence it could sometimes be useful with CI)]

Page 9, 197 exposure and short-term event independency. You did assess this by inclusion of a pre-exposure period. Would be nice to just see a plot of number of days since myocarditis patients were vaccinated? I would guess very few cases would actually be vaccinated within 1-3 weeks after an event? 

Thank you for this comment - the plot is shown below. However plots of the interval between vaccination and events can be misleading because they take no account of person-time, hence we will not include this in the paper or supplementary material. Relative incidences with varying pre exposure windows are plotted in Appendix S2, with explanation and findings; RIs change very little with pre-exposure intervals longer than 21 days which was the justification we cited for using the 21 day pre-exposure window. 

Comment: It is clear from the plot that individuals are not at all that likely to be vaccinated within 14 days of a myocarditis event. (Do not understand the comment on person-time. The study, as I understand, does not really take individual person-time into account e.g. persons are not censured after an event of at time of a vaccine-schedule that is not studied?)

Page 10, 209. A bit strange and uncommon in epidemiology to define "strong" evidence based on the p-value. :-) 

Yes this was badly worded. We have now changed this to "strong statistical evidence of an association" which was the intention behind the wording as indicated in section 4f of the protocol. 

Minor Comment: my comment was more on that "strong evidence" should more be based on the magnitude of the association, dose-response etc than on a p-value. But, it is apparent the Journal wants p-values.

Also, is there any information on the validity of these diagnoses in the registers used? This in regards to the (much) lower incidence found in your data compared to others.

Author response: The diagnoses coded as myocarditis in the SUS data set that we used were not validated against case note review (which we acknowledge in the limitations paragraph) nor were they in the Nordic study by Karlstad et al. However despite the differences in background rates in admissions for myocarditis and pericarditis between the Nordic study and our own, the findings in terms of elevated risk by vaccine product and dose in the primary series between our studies are similar. In relation to differences in background rates between the English and Nordic studies the latter used outpatient contacts for myocarditis as well as inpatient admissions. We were unable to find a breakdown in the Nordic study of the incidence rates by outpatient consultation and hospital admission so could not compare directly with the inpatient rates in our study. In Israel the Mevorach et al study only used hospital admitted cases but stated that any patient with a diagnosis of myocarditis would be admitted. Also the Israeli study used active surveillance methods in which clinicians were alerted to the potential association with vaccination and were requested to report such cases. The paragraph dealing with differences in incidence rates between studies is now expanded to provide this additional information. We do not consider that differences in background rates in different settings raises questions about the validity of the diagnoses in our study. 

Comment: 

Danish ICD codes of myocarditis, as part of the Nordic study, have been validated (Ref Sundbøll J, Adelborg K, Munch T, et al. Positive predictive value of cardiovascular diagnoses in the Danish National Patient Registry: a validation study. BMJ Open. 2016;6(11):e012832. doi:10.1136/bmjopen-2016-012832). Also, the Nordic study did not include outpatient cases: "We defined incident outcome events as the date of first hospital admission for myocarditis or pericarditis from December 27, 2020, onward. The primary outcome was a main or secondary diagnosis of myocarditis at discharge from inpatient hospital care. SARS-CoV-2 Vaccination and Myocarditis in a Nordic Cohort Study of 23 Million Residents | Cardiology | JAMA Cardiology | JAMA Network "

If the authors see no problem in large differences in background rates do they have an explanation why rates of myocarditis are much lower compared to e.g., the Nordic countries? 

I agree that it is of less importance when discussion relative measures. But, measures of absolute risk, incidence rate, excess risk are dependent on background risks. The authors do acknowledge this is the discussion. But, their explanation why they find lower rates in not really reassuring. 

Methods page 7 line 141:

"The study population comprised the resident population of 50 million individuals in England aged 12 years and older on the 31st August 2021. Dates of admission for the myocarditis or 

pericarditis were from 22nd February 2021 to 6th February 2022." Is 31 Aug 2021 the correct date? Cannot see how you can have the study pop on a later date than first outcome (22 Feb 2021)?

Page 7, line 146:

"Given the generally mild nature of vaccine-associated myocarditis [14] we analysed cases presenting in emergency care settings as well as those admitted to hospital."

Well, if the authors want to state anything on the severity of vaccine-associated myo compared to myo of other causes I would suggest reading of some more detailed references: 

Witberg G, Barda N, Hoss S, Richter I, Wiessman M, Aviv Y, et al. Myocarditis after Covid-19 vaccination in a large health care organization. N Engl J Med 2021;385: 2132-2139. 2. 

Mevorach D, Anis E, Cedar N, Bromberg M, Haas EJ, Nadir E, et al. Myocarditis after BNT162b2 mRNA vaccine against Covid-19 in Israel. N Engl J Med 2021;385: 2140-2149

Clinical outcomes of myocarditis after SARS-CoV-2 mRNA vaccination in four Nordic countries: population based cohort study | BMJ Medicine https://bmjmedicine.bmj.com/content/2/1/e000373

Page 19, line 444

"This may be attributable in part to study design as outpatient and inpatient myocarditis cases were included in the Nordic study"

This is not true. The Nordic study states: "We defined incident outcome events as the date of first hospital admission for myocarditis or pericarditis from December 27, 2020, onward. The primary outcome was a main or secondary diagnosis of myocarditis at discharge from inpatient hospital care. SARS-CoV-2 Vaccination and Myocarditis in a Nordic Cohort Study of 23 Million Residents | Cardiology | JAMA Cardiology | JAMA Network "

Page 23, line 518. Last sentence in conclusion. "The rarity and benign outcome of vaccine attributable cases is reassuring."

Well, here the authors discuss their detected incidence rates, which are (much) lower than other studies, and also discuss the severity of the outcome of which they have no supporting data except case fatality within one week. Is this sentence really needed?

Happy to see the stratification by age and sex (e.g., males 16-24) now being presented. In general, all studies on myo/peri should be stratified by sex and age as the incidence varies substantially.

[LINK]

---

## [Editor Report · Decision Letter 3]

22 May 2023

Dear Dr Miller, 

On behalf of my colleagues and the Academic Editor, Professor Rickard Ljung, I am pleased to inform you that we have agreed to publish your manuscript "Risk of myocarditis and pericarditis after a COVID-19 mRNA vaccine booster and after COVID-19 in those with and without prior SARS-CoV-2 infection; a self-controlled case series analysis in England" (PMEDICINE-D-22-03092R3) in PLOS Medicine.

Prior to publication, please remove the redundant opening parenthesis in S3 Table '(1.24, (6.85)' from the penultimate column, row 5.

PRESS

Best wishes, 

Pippa

Philippa Dodd, MBBS MRCP PhD 

PLOS Medicine